# Characterization and Function of the 1-Deoxy-D-xylose-5-Phosphate Synthase (DXS) Gene Related to Terpenoid Synthesis in *Pinus massoniana*

**DOI:** 10.3390/ijms22020848

**Published:** 2021-01-15

**Authors:** Rong Li, Peizhen Chen, Lingzhi Zhu, Fan Wu, Yu Chen, Peihuang Zhu, Kongshu Ji

**Affiliations:** Key Laboratory of Forest Genetics & Biotechnology of Ministry of Education, Nanjing Forestry University, Nanjing 210037, China; hblxylr@126.com (R.L.); pei_jane@126.com (P.C.); lingzhi_zhu@126.com (L.Z.); eiknarf@126.com (F.W.); Chenyu516106@126.com (Y.C.); phzhu@njfu.edu.cn (P.Z.)

**Keywords:** *Pinus massoniana*, DXS, expression pattern, terpenoid, prokaryotic expression, chlorophyll

## Abstract

In the methyl-D-erythritol-4-phosphate (MEP) pathway, 1-deoxy-D-xylose-5-phosphate synthase (DXS) is considered the key enzyme for the biosynthesis of terpenoids. In this study, *PmDXS* (MK970590) was isolated from *Pinus massoniana*. Bioinformatics analysis revealed homology of MK970590 with DXS proteins from other species. Relative expression analysis suggested that *PmDXS* expression was higher in roots than in other plant parts, and the treatment of *P. massoniana* seedlings with mechanical injury via 15% polyethylene glycol 6000, 10 mM H_2_O_2_, 50 μM ethephon (ETH), 10 mM methyl jasmonate (MeJA), and 1 mM salicylic acid (SA) resulted in an increased expression of *PmDXS*. pET28a-*PmDXS* was expressed in *Escherichia coli* TransB (DE3) cells, and stress analysis showed that the recombinant protein was involved in resistance to NaCl and drought stresses. The subcellular localization of *PmDXS* was in the chloroplast. We also cloned a full-length 1024 bp *PmDXS* promoter. *GUS* expression was observed in *Nicotiana benthamiana* roots, stems, and leaves. *PmDXS* overexpression significantly increased carotenoid, chlorophyll a, and chlorophyll b contents and DXS enzyme activity, suggesting that *DXS* is important in isoprenoid biosynthesis. This study provides a theoretical basis for molecular breeding for terpene synthesis regulation and resistance.

## 1. Introduction

Rosin is a complex resinous mixture of terpenes secreted by conifers that is characterized by a special smell and is widely used as a spice, medicine, and insecticide [1]. There are two biosynthetic pathways for rosin: the mevalonate pathway (MVA) in the cytoplasm and the methyl-D-erythritol-4-phosphate pathway (MEP) in plastids. In addition, the main products of the former pathway are monoterpenoids and diterpenes, while the latter pathway can synthesize sesquiterpenes [2]. Rosin contains turpentine; rosin is mainly synthesized from nonvolatile diterpenes (C20), while turpentine is mainly synthesized from volatile monoterpenes (C10) and sesquiterpenes (C15). 1-Deoxy-d-xylose-5-phosphate synthase (DXS) is the first key enzyme in the MEP pathway. DXS uses pyruvate and *D*-glyceraldehyde-3-phosphate (D-GAP) as substrates and produces 1-deoxy-d-xylose-5-phosphate (DXP) under the action of thiamine pyrophosphate (TPP) [3]. Although the precursor materials, biochemical reaction steps, and catalysis enzymes of the MVA and MEP pathways are completely different, the final products of both are IPP and DMAPP [4].

Overexpression of the *DXS* gene can increase the production of terpenoids in host plants, thus indirectly improving the resilience of these plants [5]. Compared with those of wild-type *Arabidopsis thaliana*, plants overexpressing the *StDXS1* gene of potato showed gradual accumulation of carotenoids and chlorophyll, and the levels of abscisic acid and gibberellin were lower, proportional to the increase in the germination rate of the overexpression plants. It was also found that the expression of *GGPPS*, a downstream gene of the MEP pathway, was upregulated. In addition, in terms of disease resistance, StDXS1 was found to be related to the level of isoprenoid-induced resistance of the plant against the potato late blight pathogen in transgenic lines [6]. However, some studies have shown that the ABA (Abscisic Acid) content in *A. thaliana* overexpressing *DXS* is significantly increased compared to that of the wild type [7]. In addition, overexpression of *GrDXS* in rose-scented geranium and *Withania somnifera* leads to an increase in the content of essential oils, and the latter leads to an increase in the content of isoprene [8]. This is consistent with the results of overexpression of bacterial *DXS* in tomato [9]. The level of DXS enzyme activity is related to the accumulation of terpenoids. With RT-qPCR detection, the expression level of most genes in the MEP pathway of *Zanthoxylum bungeanum* increased, but the expression level of *ZbDXS* reached the highest level among the genes, which was 12 times that of the control, while the rate-limiting enzyme HMGR of the MVA pathway was only 3 times higher than that of the control [10]. The DXS enzyme is the rate-limiting enzyme in the MEP pathway and plays a key role in the synthesis of secondary metabolites and plant stress.

Masson pine (*Pinus massoniana*) is not only the most widely distributed timber species in southern China but also the main lipid-producing tree species, and approximately 90% of pine resin in China is produced from Masson pine [11]. To date, more than 70 terpene synthase enzymes have been identified in pines [12], but little information is available on the key enzyme gene in the process of terpenoid synthesis of Masson pine 1-deoxy-D-ketose-5-phosphate synthase (DXS). In this study, we aimed to reveal the key functions of *PmDXS* in Masson pine stress resistance through *PmDXS* gene cloning, tissue- and stress-specific expression pattern, subcellular localization, prokaryotic expression, and transient expression analyses.

## 2. Materials and Methods

### 2.1. Plant Materials

The experimental materials were 2-year-old potted Masson pine seedlings from the seed orchard of the Baisha state-owned forest farm, Shanghang, Fujian Province, China, and 15-year-old Masson pines on the campus of Nanjing Forestry University. Wild-type *Nicotiana benthamiana* and *A. thaliana* ecotype Columbia (Col-0) were used in this study.

### 2.2. Cloning of the mRNA Sequence of the PmDXS Gene

According to the SMARTer^®^ RACE cDNA Amplification Kit and the 3′Full RACE Core Set with PrimeScript RTase (TaKaRa, Beijing, China) kit instructions, the 5′ and 3′ ends of the complete first strand of Masson pine cDNA were synthesized. After searching the *DXS* sequences of different plants in NCBI (National Center for Biotechnology Information), comparing with our laboratory’s transcriptome database, we screened out the target mRNA and designed primers for the middle fragment of *PmDXS*, which were named PmDXS-Mid (Table 1). PCR was performed with the cDNA of Masson pine seedlings as a template. The reaction system had a 50 μL total volume, including cDNA (50 ng/µL) 2 µL, PmDXS—MidF 1 µL, PmDXS—MidR 1 µL, 5 × GXL Buffer (Mg^2+^Plus) 10 µL, DNTP Mixture 5 µL, GXL enzyme 1 µL, and ddH_2_O to reach 50 µL; the reaction program was as follows: 98 °C for 3 min, followed by 98 °C for 15 s, 58 °C for 15 s, and 72 °C for 2 min for 35 cycles, with a final extension at 72 °C for 7 min and held at 4 °C + ∞.

Taking the sequence of the intermediate fragment of *PmDXS* obtained by sequencing as a reference, specific primers PmDXS 5′RACE Outer, PmDXS 5′RACE Inner, PmDXS 3′RACE Outer, and PmDXS 3′RACE Inner were designed (Table 1). Using the prepared RACE cDNA as a template, the sequences at both ends were obtained by nest-PCR amplification. The instructions of the SMARTer^®^ RACE cDNA Amplification Kit and 3′Full RACE Core Set with PrimeScript RTase (TaKaRa) kits were followed for the reaction steps. To verify the correctness of the open reading frame (ORF) region, we searched the ORF region through the Open Reading Frame Finder (ORF Finder) on the NCBI website and designed primers at both ends of the ORF. DXS-ORF primers (Table 1) and Masson pine cDNA were used as templates for PCR (Polymerase Chain Reaction) amplification. After gel electrophoresis detection, we recovered the products from the gel according to the manufacturer’s instructions (Yeasen Biotech, Shanghai, China). The recovered product was ligated into pEASY-Blunt vector (Transgen Biotech, Beijing, China), transformed into Escherichia coli Trans1-T1 (Transgen Biotech, Beijing, China), and cultured at 37 °C overnight. Blue and white spots were screened with the LB (Lysogeny Broth) solid medium containing X-Gal and IPTG (isopropyl β-D-thiogalactoside). Then a single white colony was selected for PCR detection, and the recombinant plasmid was sequenced by Jie Li Company (Shanghai, China).

### 2.3. Bioinformatics Analysis of PmDXS

The ExPASy (http://web.expasy.org/protparam/) online software was used to analyze the basic physical and chemical properties of amino acid sequences. SOPMA online software (https://npsa-prabi.ibcp.fr/cgi-bin/npsa_automat.pl?page=npsa_sopma.html) was used to analyze and predict the secondary structures of proteins. DNAMAN and Jalview software were used to compare the amino acid sequence of PmDXS with those from other species for homology [13]. TMHMM Server version 2.0 was used to predict the transmembrane domain (http://www.cbs.dtu.dk/services/TMHMM). MEGA X [14], Evolview (https://www.evolgenius.info/evolview/) [15] and iTOL version 5 (https://itol.embl.de/) were used to build a phylogenetic tree. The TargetP1.1 Server (http://www.cbs.dtu.dk/services/TargetP/) and Cell-PLoc version 2.0 (http://www.csbio.sjtu.edu.cn/bioinf/Cell-PLoc-2/) [16] were used for subcellular localization analysis of the PmDXS protein.

### 2.4. qRT-PCR

We used an RNA extraction kit (DP441, Tiangen Biotech, Beijing, China) to extract RNA from different tissues of 15-year-old Masson pine and two-year-old robust seedlings of Masson pine under different treatments. Different tissues included roots (R), young stems (NS), old stems (OS), young leaves (NL), old leaves (OL), flowers (F), xylem (M), and phloem (P). Different treatments included abiotic stress and hormone treatment, including mechanical damage, 15% polyethylene glycol (PEG 6000), 10 mM H_2_O_2_, 50 μM ethephon (ETH), 10 mM methyl jasmonate (MeJA), and 1 mM salicylic acid (SA) [17]. The mechanical damage treatment method was performed by cutting the upper half of the pine needles, the osmotic stress was induced by soaking the plants in PEG 6000, and the remaining abiotic stress and hormone treatments were applied by spraying the plant surfaces. The needles of Masson pine were collected every 0, 3, 6, 12, 24, and 48 h, immediately placed in liquid nitrogen, and stored at −80 °C. For all of the above treatments, 0 h was used as the control group without any treatment. There were 3 biological replicates for each sample, 3 strains per replicate, 3 technical replicates, and 3 strains per replicate. A Q-RT cDNA reverse transcription kit (Yeasen Biotech, Shanghai, China) was used to synthesize cDNA. Primer 5.0 was used to design the real-time quantitative PCR primer Q-DXS for the *PmDXS* gene (Table 1), and *Actin2* (GenBank accession number: KM496525.1) [18] was used as the reference gene. cDNA diluted 20-fold was used as the template for fluorescence quantitative PCR. The Hieff UNICON Universal Blue qPCR SYBGreen Master Mix instruction manual was used for the reaction system and procedures. The 2^−ΔΔCT^ method was used to calculate the relative expression. We used Microsoft Office Excel 2019, GraphPad Prism version 8.0, Adobe Illustrator CS6, and IBM SPSS Statistics version 25.0 for data analysis and image production.

### 2.5. Construction and Transient Expression of Subcellular Localization Vector

We constructed the pCAMBIA1302 vector containing the GFP (Green fluorescent protein) fluorescence signal, constructed the PmDXS::GFP fusion vector, designed the primer 1302-DXS (Table 1) containing the restriction sites of *Nco* Ⅰ and *Spe* Ⅰ, transformed the *Agrobacterium* competent GV3101, and injected it with a syringe Ben’s tobacco [19,20]. A laser confocal microscope (LSM710, Zeiss, Germany) was used to observe the GFP signal of the epidermal cells of *N. benthamiana*, and the image was observed by ZEN Imaging version 2.36 software.

### 2.6. Construction of the Prokaryotic Expression Vector

We used pET28a as the prokaryotic expression vector, selected *EcoR* I and *Hind* III restriction enzymes as the restriction enzymes and used SnapGene software to design the prokaryotic expression vector primer 28a-DXS, which contained restriction sites (Table 1), referring to the One Step Instructions for the Cloning Kit (Vazyme Biotech, Nanjing, China) for recombination reactions. The recombinant vector pET28a-*PmDXS* was transformed into the host strain TransB (DE3) (Transgen Biotech, Beijing, China).

### 2.7. Production of the Target Protein and Stress Treatments

In this study, temperature (37 °C and 16 °C) and IPTG were used to induce stress. The positive expression strain TransB (DE3) (1 mL) was inoculated into 50 mL after shaking overnight. Then, the positive expression strain TransB (DE3) (1:50) was inoculated into 50 mL of LB liquid medium containing Kan (50 mg/mL) at 220 rpm and 37 °C. When OD_600_ = 0.6~0.8, 0 mM, 0.5 mM, or 1.0 mM IPTG were added, and the mixture was shaken at 37 °C for 4 h, at 16 °C for 16 h, or at 4 C° at 5000 rpm·min^−1^ for 15 min, and the bacteria were collected. The bacteria were washed with precooled 1× PBS buffer, resuspended, and lysed by boiling (100 °C for 2 min) to obtain the whole bacterial sample; then, the samples were centrifuged for 5 min at 4 °C and 5000 rpm·min^−1^, and the supernatant and pellet samples were collected. At the same time, to optimize the induction expression system of pET28a-PmDXS (TransB), the induction temperature was set at 37 °C, the shaking time was 4 h, and the pET28a empty vector bacterial solution with 1 mM IPTG was used as the control group. The concentration gradient was 0, 0.1, 0.3, 0.5, 1.0, 2.0 mM IPTG pET28a-PmDXS recombinant protein as the test group, and the optimal concentration of IPTG was determined. The expression of the recombinant protein was observed in the supernatant sample, and finally, 1×SDS-PAGE (ten Sodium dialkyl sulfate-polyacrylamide, 10% separation gel, 5% concentrated gel) electrophoresis was performed to detect the target band [19].

We used LB liquid to dilute the expressed bacterial solution at 16 °C to 10^0^, 10^−1^, 10^−2^, 10^−3^, 10^−4^, and 10^−5^ times and added 5 μL of each to 400 mM NaCl, 800 mM D-mannitol, pH 5.0 (HCl), and pH 9.0 (NaOH) solid medium (LB is the control group) [20]. The droplet diameter was approximately 0.5 cm, and the bacterial liquid was blown dry in an ultraclean workbench and incubated at 37 °C overnight to observe the growth of the colony.

### 2.8. Promoter Cloning and Expression Analysis

We extracted 15-year-old Masson pine genomic DNA (DP320, Tiangen Biotech, Beijing, China), designed two reverse primers: GSP1 and GSP2, and used Tail-PCR for 3 rounds of amplification. Takara’s Genome Walking Kit instructions were followed for specific methods. Based on the sequencing results, we designed the full-length specific primer PmDXS-Pro for the *PmDXS* promoter (Table 1) to amplify the product of the full-length promoter. The online software PlantCARE (http://bioinformatics.psb.ugent.be/webtools/plantcare/html/) was used to predict and analyze the cis-acting promoter elements.

In this study, pBI121 containing the *GUS* (β-glucuronidase) reporter gene was used as a transient expression vector, and the pBI121-ProDXS primer containing the *Xba* I and *Hind* III double restriction sites was designed with SnapGene software (Table 1). The recombined pBI121-PmDXS pro::GUS transient expression vector and pBI121 empty plasmid vector were transformed into *Agrobacterium* GV3101 cells, and the *N. benthamiana* tissue cultured plants (20-day seedling age) were transiently transformed by the *Agrobacterium*-mediated method [21]. After incubating at 28 °C for 2 days in the dark, the roots, stems, and leaves were cut and decolorized with 75% ethanol. Finally, GUS staining was performed under a stereomicroscope (SZX16, OLYMPUS, Tokyo, Japan) and an stereo fluorescence microscope (M205FA, Leica, Wetzlar, Germany).

### 2.9. Genetic Transformation and Pigment Content Determination of Overexpression Plants

The *A. thaliana* ecotype Columbia (Col-0) seeds were soaked in 75% alcohol for 3 min, 20% sodium hypochlorite for 2 min, repeatedly rinsed with sterile water 4–5 times, and placed on filter paper to fully dry. The seeds were spotted on 1/2 MS solid medium (1/2 MS salt, 1% sucrose, 0.6% agar, pH 5.8) [22]. The seeds were vernalized for 3 days, placed in a 24 °C light incubator (16 h light/8 h dark) for one week, and then transferred to the soil to continue growing. The pCAMBIA1302-35S::*PmDXS*::GFP overexpression vector constructed in 1.5.7 was transformed into *Agrobacterium* EHA105. In this experiment, the floral dip method was used to transform wild-type A. thaliana plants [23] genetically. Transgenic plants were screened on 1/2 MS (Murashige and Skoog) medium containing 1% sucrose and 25 µg/mL Hyg (hygromycin). Two weeks later, after identification by genomic PCR, the plants confirmed to be transgenic were transplanted into pots for continued growth until the T2 generation seeds were harvested for subsequent experiments.

Twelve transgenic lines with high expression levels were selected, and the four indexes of carotenoid, chlorophyll a, and chlorophyll b, and DXS enzyme activity were determined by Huding Biotech (Shanghai, China).

## 3. Results

### 3.1. Molecular Cloning and Sequence Analysis of PmDXS

According to the Masson pine transcriptome data [24], the *DXS* gene was cloned and spliced to obtain a full-length sequence of 2513 bp (GenBank accession number: MK970590), including the middle 1709 bp fragment (Figure 1a), 5′RACE 991 bp fragment (Figure 1b), 3′RACE 333 bp fragment (Figure 1b), and 2223 bp ORF region (Figure 1c). In addition, the *DXS* gene can encode a 740 amino acid protein. The secondary structure of the protein predicted by SOPMA mostly includes random coils, followed by α-helices and β-turns. The physical and chemical properties of the amino acids of the *PmDXS* gene were analyzed by ProtParam online software. The isoelectric point (pI) was 8.54, the number of total negatively charged residues (Asp+Glu) was 76, and the number of total positively charged residues (Arg+Lys) was 82. The numbers of 20 species and their proportions are shown in Figure 1d. Similar to DXS proteins in other plants, the PmDXS protein is a hydrophilic protein, and it does not contain a transmembrane domain and signal peptide.

A conserved domain analysis of predicted amino acids was performed on the NCBI website (Figure 1e), and the results showed that the protein contained the core sequences of the TPP_enzyme and TPP_enzyme_PYR superfamilies. Between the 73rd and 736th amino acids, there is a multifunctional domain of PLN02582, which is a characteristic sequence of 1-deoxyxylulose-5-phosphate synthase. Therefore, the protein encoded by *PmDXS* belongs to the family of 1-deoxyd-d-xylose-5-phosphate synthases. The PmDXS protein was analyzed and predicted with Jalview software, and it was found that it contained the ammonium disulfate binding site GDGG(X)8E(X)4A(X)11NDN (marked with red dotted line) and a transketolase domain DRAGX28PXD (marked with solid red line). After comparing the amino acid sequence of PmDXS with the DXS amino acid sequences of the other 4 plant species, it was found that the similarity was more than 90%, and the similarity reached 99% with those of *P. kesiya* and *P. densiflora* (Figure 1f). TargetP1.1 Server and Cell-PLoc 2.0 online software were used to analyze the subcellular location of the DXS protein of Masson pine, and the results showed that the protein is most likely located in the chloroplast. Analysis of the DXS amino acid sequence of PmDXS and DXS proteins from 18 other species of plants showed that gymnosperm DXSs clustered together and were closely related in *P. densiflora*, Simao pine (*P. kesiya*), and *Picea abies* (Figure 1g).

### 3.2. Tissue-Specific Expression Analysis of PmDXS

qRT-PCR technology was used to analyze flower (F), young leaf (NL), old leaf (OL), young stem (NS), old stem (OS), root (R), xylem (M), and phloem (P) expression. The results showed that the gene was expressed in all organs. The expression levels in the flower, xylem, and phloem were lower than those in young leaf tissues, with 24.3%, 15.6%, and 31.1% of those in young leaves, respectively. The expression level in roots was the highest among the plant tissues, being 3.2 times higher than those in other tissues (Figure 2). Thus, *PmDXS* is mainly expressed in the roots of Masson pine.

### 3.3. Analysis of PmDXS Expression Pattern under Abiotic Stress

Two-year-old Masson pine seedlings were treated with six abiotic stresses, and qRT-PCR analysis showed that the relative expression levels of PmDXS under the conditions of mechanical injury, 15% PEG 6000, 10 mM H_2_O_2_, 50 µM ETH, 10 mM MeJA, and 1 mM SA were significantly higher than those of the control group (Figure 3a–f). The expression level of *PmDXS* was significantly higher under H_2_O_2_ stress than that of the control group and the other stress groups (Figure 3c). When treated with PEG 6000 (15%), MeJA (10 mM), and H_2_O_2_ (10 mM) for 6 h, the expression of *PmDXS* was the highest, being 2.01 times (Figure 3e), 2.08 times (Figure 3a), and 3.31 times (Figure 3c) that of the control, differences which were significant. Under ETH stress, the expression of *PmDXS* first increased and then decreased until the highest expression level was reached at 12 h, which was 1.14 times that of the control group (Figure 3b). When SA (1 mM) stress was applied for 3 h, *PmDXS* expression was the highest, which was 3.39 times that of the control group (Figure 3d). Under mechanical injury, the expression of *PmDXS* showed a trend of increasing first and then decreasing, reaching the highest values at 3 h and 6 h, which were 6.63 times and 7.3 times those of the control (0 h), respectively (Figure 3f). The above results indicated that *PmDXS* participates in the abiotic stress response mechanism and plays a role in the stress response of Masson pine.

### 3.4. Analysis of PmDXS Expression Pattern under Abiotic Stress

#### 3.4.1. Induced Expression Analysis of Recombinant Protein

The positive prokaryotic expression strain (TransB/pET28a-PmDXS) was mixed with 0 mM, 0.5 mM, 1.0 mM IPTG, and the expression was induced by shaking at 37 °C for 4 h and 16 °C for 16 h. After SDS-PAGE analysis, the *E. coli* with the pET28a empty plasmid vector was induced with different IPTG concentrations at 65–100 KDa, and no specific protein appeared. However, the *E. coli* containing pET28a-PmDXS was significantly different after induction with different concentrations of IPTG compared with the control group without IPTG. Obvious thick bands were seen at 70 kDa in the whole bacterial samples and supernatant samples at 37 °C and 16 °C, indicating the appearance of specific protein bands. This result was consistent with the expected PmDXS protein size of 70.4 kDa; that is, the *PmDXS* gene was expressed in *E. coli* (Figure 4).

#### 3.4.2. Experimental Analysis of Recombinant Protein under Different Stresses

The expression difference between the recombinant bacteria and the empty vector under stress conditions was observed through the drop plate test (Figure 5), and it was found that the growth state of the empty vector (control) on the LB solid medium was consistent with the growth state of the recombinant bacteria, while under treatment with 400 mM NaCl, 800 mM D-mannitol, pH 5.0 (HCl), or pH 9.0 (NaOH), the number of recombinant bacteria was significantly higher than that of control bacteria.

### 3.5. Subcellular Localization of PmDXS

The *PmDXS*::GFP fusion expression vector was injected into young tobacco leaves (20-day seedling age) by the transient transformation method. After culturing in dark light for 2 days in an artificial climate incubator, 0.5 × 0.5 cm tobacco leaf samples were taken and laid flat on a glass slide to be pressed. The expression was observed under a confocal laser microscope. The green fluorescence was concentrated in the chloroplasts of leaf epidermal cells and was more obvious in the guard cells (Figure 6). Therefore, the PmDXS protein can be considered a plastid protein.

### 3.6. Function Analysis of PmDXS Promoter

Using Masson pine genomic DNA as a template, a 1024 bp promoter sequence was obtained after PCR amplification (Figure 7a,b). After the sequencing results were compared, and no base mutation site was found, indicating that the specific fragment cloned was a promoter sequence upstream of the transcription start site, named PmDXS-Pro. The PlantCARE online tool was used to predict and analyze the cis-acting elements of the obtained promoters. The sequence contained 10 types of cis-acting elements, with typical core elements TATA-box (22 sites) and CAAT-box (24 sites) of eukaryotic promoters, as well as elements involved in the ABA response (ABRE); a light-responsive element (G-Box); a cis-acting element related to the salicylic acid reaction (TCA-element); a cis-acting regulatory element involved in the MeJA reaction (TGACG-motif); and a cis-regulatory element of the auxin response (AuxRR-core) (Figure 7c). It is speculated that PmDXS-Pro *expression* may be regulated by the photoperiod and plant hormone signals. In addition, the presence of the MYB binding site MRE indicates that the transcription factors related to Masson pine stress resistance may play a role in regulating the expression of *PmDXS*.

Wild-type *N. benthamiana* was used as a negative control, and *N. benthamiana* infected with pBI121:GUS empty vector *Agrobacterium* liquid was used as a positive control. The roots, stems, and leaves of *N. benthamiana* were infected with *Agrobacterium* for GUS staining. The results showed that the wild tobacco infected with the negative control was not stained and that infected with the positive control showed an obvious blue color. The tobacco infected by the test group showed a lighter blue response, indicating that the cloned *PmDXS* has a promoter function and can also drive the expression of the downstream *GUS* under noninducible conditions (Figure 7d).

### 3.7. Chlorophyll and Carotenoid Contents and DXS Enzyme Activity in Transformed and Wild-Type A. thaliana

The total chlorophyll includes chlorophyll a and chlorophyll b. Compared with the wild-type control, the chlorophyll a content was highest in L12, L3, and L6 transgenic plants, increasing by 1.7, 1.56, 1.55 times, respectively, while the L4, L9, and L5 strains showed the highest chlorophyll b contents, which were 2.06, 1.77, 1.77 times that of the control, respectively. The chlorophyll a and b contents of other strains increased by more than 11% compared with the control. The carotenoid content in the transgenic line was significantly higher than that in the wild type. L2 reached 233.06 pg/mL, which was 1.60 times that of the control, while L7 had the lowest value, which was 1.22 times that of the control. The rest of the lines all increased by 12%.

DXS enzyme activity analysis showed that the transgenic lines L8 and L2 had the highest expression level of 194.64 U/L, which was 1.80 times that of the control, followed by the L4 line at 184.83 U/L, which was 1.71 times that of the wild type, and the L1 line had the lowest activity at 1.16 times that of the control. The DXS enzyme activity of the other strains increased by more than 12% compared with the control (Figure 8).

## 4. Discussion

### 4.1. Characteristic Analysis of PmDXS

Plant terpenoids play an important role in the growth and development of plants, and they are widely used in new antibiotics, new herbicides, and antimalarial drugs [25]. 1-Deoxy-D-xylulose-5-phosphate synthase is a key regulatory site for downstream products of the MEP pathway and guides the synthesis of terpenoids. The *DXS* gene family has been widely studied, and different plants contain different types of *DXS* genes. *P. abies* has three *DXS* genes, *PaDXS1* was constitutively expressed, and its transcription level was not affected by external damage and fungal infection, while *PaDXS2A* and *PaDXS2B* were vulnerable to stresses [26]. Among the four *DXS* genes cloned from *Artemisia carvifolia*, *AaDXS1* belongs to the *DXS* Ⅰ type, *AaDXS2* and *AaDXS3* belong to the *DXS* Ⅱ type, and *AaDXS4* belongs to the *DXS* Ⅲ type. Laser confocal microscopy shows that the DXSs are all located in chloroplasts [27]. Simao pine also contains three *DXS* genes: *PkDXS1*, *PkDXS2*, and *PkDXS3*; the latter two belong to *DXS* Ⅱ type and are especially affected by physical damage but had less impact on *PkDXS1* [28].

This experiment cloned the *PmDXS* gene of Masson pine for the first time. From the perspective of the molecular evolutionary tree, the *PmDXS* gene clustered with those of gymnosperms, such as *PkDXS2* (AIY22670.1), *PdDXS2* (ACC54554), *PaDXS2* (ABS50519.1), and *GbDXS2* (AAR95699.1). Together, these results show that the *PmDXS* gene belongs to the *DXS* II group from gymnosperms and is usually associated with encoding plant secondary metabolites. In addition, combined with the transcriptome data from young Masson pine, it can be speculated that Masson pine may contain at least two *DXS* genes. This is consistent with the results of multiple *DXS* genes in the above studies. The predicted amino acid conserved domains indicated that the protein contained the pyrimidine binding domain (TPP_PYR_DXS-like), the C-terminal transketolase domain (Transketolase-C), and the characteristic sequence of the PLN02582 functional domain. In addition, the PmDXS protein contained an ammonium disulfate binding site and a transketolase domain, which is consistent with the results of many studies [6,29,30]. Therefore, it is speculated that the protein belongs to the family of 1-deoxyxylulose-5-phosphate synthases. The subcellular localization test found that PmDXS was located in the chloroplast, which is consistent with the location of the MEP pathway in the plastid [6,31].

### 4.2. Analysis of PmDXS Organization and Induced Expression Pattern

The expression of the *DXS* gene may also be regulated by exogenous factors. The expression of the plant *DXS* gene family is tissue-specific, and the expression of different *DXS* genes varies significantly in different plants and different organs. The relative expression of *TcDXS* in different tissues of *Taxus chinensis* showed a trend of tender petioles > leaves > barks > roots > stems [32]. *AlDXS* has the highest expression level in leaves, followed by flowers, and low expression in rhizomes and roots [33]. The analysis of the expression patterns [34] in the model plants *N. tabacum*, *Glycine max*, and *Solanum lycopersicum* showed that *DXS1* was less expressed in the root system, while *DXS2* was most highly expressed in the roots, which was similar to the tissue expression results in this experiment. The expression level of *PmDXS* in roots was significantly higher than that in other tissues, indicating that with the development of plant organs to different degrees, the transcription level of the secondary metabolism of this gene was also different. After treating *Dendrobium officinale* with ABA, SA, and brassinolide (BR), the expression of *DXS* in the protocorm was significantly increased [35]. *AsDXS1* in *Aquilaria sinensis* can be significantly stimulated by mechanical, chemical, and H_2_O_2_ stresses, while *AsDXS2* and *AsDXS3* only responded to chemical and mechanical treatments, respectively. Therefore, these three genes were all regulated by MeJA, but the time points of peak expression were different [30]. The four *DXS* genes in *A. annua* had different responses to damage and MeJA stress treatments. *AaDXS2* and *AaDXS3* were induced by MeJA, and the expression of *AaDXS3* was significant. In addition, only *AaDXS1* was upregulated after mechanical damage induction [27]. This study also confirmed that treatment with MeJA (10 mM), H_2_O_2_ (10 mM), SA (1 mM), ETH, and PEG6000 (15%) induced upregulated expression of *PmDXS*. In addition, *PmDXS* showed a trend of first rising, then falling, and then rising after mechanical damage treatment. *PmDXS* was upregulated and expressed in a short period of time, suggesting that this gene may be involved in the synthesis of terpenoids under adverse stress, thereby resisting the damage to Masson pine caused by adverse factors. Therefore, it can be speculated that the *DXS* gene has temporal and spatial specificity, which can increase the expression of *PmDXS* at the transcription level, thereby affecting the synthesis of terpenoids and improving stress resistance. This agrees with research on the *PdDXS* gene of *P. densiflora* and the *GbDXS* gene of *Ginkgo biloba* [36,37].

### 4.3. Relationship between PmDXS Prokaryotic Expression and the Stress Response

Prokaryotic expression technology has the advantages of a short cycle and simple operation. The *E. coli* expression host system can be used to initially identify genes after constructing the prokaryotic expression vector [20]. The *Lavandula angustifolia* pET28a-DXS prokaryotic expression vector has been successfully expressed in Transetta (DE3) [38]. The cumulative amount of the paclitaxel recombinant protein pET28a-TcDXS/Rosetta (DE3) is different depending on the induction time, and the maximum induction was achieved with 4 h at 37 °C [39]. To further study the role of *PmDXS* in stress resistance, this experiment successfully induced the TransB/pET28a-*PmDXS* prokaryotic expression strain. In addition, the growth status of the recombinant protein on the LB solid plates treated with 400 mM NaCl, 800 mM D-mannitol, pH 5.0, and pH 9.0 was significantly better than that of the control strain. Therefore, *PmDXS* can promote the growth of *E. coli* and simultaneously exhibit a positive regulatory effect under salt and drought stress. We speculate that this gene plays a role in the growth of Masson pine and the process of adverse stress. This is similar to the findings of *Populus trichocarpa* [5].

### 4.4. Transient Expression of the PmDXS Promoter

Promoters play an important role in the regulation of plant transcription levels. The promoters of eukaryotes are mostly composed of core promoter regions and upstream promoter elements [40]. In this study, the chromosome walking method was used to clone the *PmDXS* upstream promoter sequence. The analysis of the cis-acting elements of the promoter showed that the *PmDXS* promoter region contained a total of 10 regulatory elements, which were divided into four functions: transcriptional regulation, stress response, light signal transduction, and plant hormone response. These results indicate that the expression of *PmDXS* may be regulated by light and hormones and has an important regulatory effect on plant growth and development. In addition, after the transformation of tobacco by the transient transformation method, it was found that *PmDXS* may drive the expression of the *GUS* reporter gene; that is, *GUS* expression was detected in roots, stems, and leaves. This is similar to the research results of *Gossypium hirsutum* [41] and *Elaeis guineensis* [42]. The above results showed that the promoter sequence of *PmDXS* had typical plant promoter characteristics and tissue specificity. The study of the *PmDXS* promoter has an important reference value for revealing the function of *PmDXS* and, at the same time, provides a theoretical basis for research on Masson pine stress.

### 4.5. Function Analysis of Overexpression Plants

As the first rate-limiting enzyme of the MEP pathway, the DXS enzyme plays an extremely important role in the regulation of the terpenoid synthesis pathway. Numerous studies have shown that the overexpression of the *DXS* gene is related to an increase in terpenoid content. Chlorophyll, carotenoids, abscisic acid, and gibberellin in *A. thaliana* overexpressing *AtDXS* accumulated to varying degrees, and their germination rate was different from that in wild-type *A. thaliana* [31]. In addition, some researchers compared plants that silently expressed *NtDXS* with control plants; the total chlorophyll content of the leaves was reduced by 45.3%, and the carotenoid content was reduced by 40.2%. This confirmed the importance of *DXS* in the MEP pathway to the synthesis of aroma precursor products [25]. After antisense expression of the *PpDXS1* gene in *Poa pratensis*, the expression of genes related to chlorophyll biosynthesis in the transgenic lines was upregulated [43]. In this study, we transferred *PmDXS* into *A. thaliana*, which resulted in an increase in chlorophyll and carotenoid contents and DXS enzyme activity in the transgenic lines. This is similar to the findings that the chlorophyll content of *A. thaliana* increased after overexpression of *StDXS1* [6].

## 5. Conclusions

In this study, we cloned a full-length *PmDXS* and part of its promoter from *P. massoniana*. We also performed bioinformatics analysis on this gene, and the molecular evolutionary tree showed that it belongs to the *DXS* Ⅱ type. The subcellular localization results indicated that the PmDXS protein is located in the chloroplast. We assessed the expression patterns of the gene under abiotic stress and the expression of the recombinant protein under different stresses, which showed a positive correlation with stress resistance in Masson pine. The *PmDXS* promoter region has multiple cis-functional elements involved in stress and hormone responses that can drive *GUS* activity in different tissues of tobacco. This study also analyzed the changes in the contents of secondary metabolites in transgenic and wild-type plants. Overexpression of *PmDXS* increased the chlorophyll content and DXS enzyme activity. Our findings will aid the further study of the function of the *PmDXS* gene and provide a reference for molecular resistance breeding in *P. massoniana*.

## Figures and Tables

**Figure 1 ijms-22-00848-f001:**
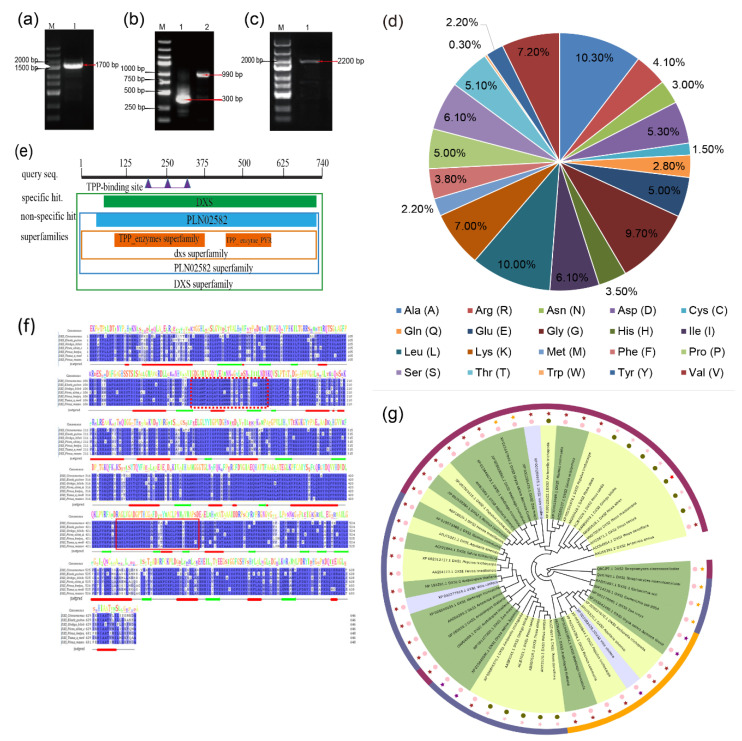
Cloning and sequence analysis of *PmDXS*. (**a**) The cloning of the intermediate fragment (1. PCR amplification production of the intermediate fragment). (**b**) The cloning of the two end sequences (1. PCR amplification product of the 3′ RACE fragment; PCR amplification product of the 5′ RACE fragment). (**c**) The cloning of the ORF fragment (1. PCR amplification product of the ORF fragment). (**d**) The number and proportion of amino acids in the PmDXS protein. (**e**) Prediction of conservative protein sequences encoded by *PmDXS*. (**f**) Amino acid sequence alignment of *PmDXS* from *P. massoniana* and other plants. Thiamine diphosphate binding sites are marked by dotted box lines, and transketolase domains are marked by solid box lines. The “jnetpred” in the picture represents the prediction result of the secondary structure of DXS protein: α-helices are represented by red tubes, and β-sheets are represented by green arrows. (*Cinnamomum micranthum* RWR84357.1, *Elaeis guineensis* XP_010933574.1, *Ginkgo biloba* AAR95699.1, *Picea abies* ABS50520.1, *Pinus kesiya* AIY22671.1, *Pinus massoniana* MK970590, *Taxus × media* AAS89342.1). (**g**) Molecular phylogenetic tree of *PmDXS.* Background colors: dark green represents herbaceous plants, light green represents woody plants, light blue represents vines. Colors of outer bars: dark blue represents *DXS* type I, purple–red represents *DXS* type II, orange represents *DXS* III Type. The color of the stars: brown represents dicots in angiosperms, pink represents gymnosperms, purple represents monocots in angiosperms, orange represents fungi. Color of dots: pink represents angiosperms, green represents gymnosperms plant. (*Amborella trichopoda* DXS1 XP_006855370.1, DXS2 XP_020523522.1, DXS3 XP_020529570.1, *Aquilaria sinensis* DXS1 AFU75321.1, DXS2 AHI62962.1, DXS3 AFU75320, *Arabidopsis thaliana* DXS1 NP850620.2, DXS1-2 NP193291.1, DXS2 OAP04569.1, DXS3 AED91669.1, *Ginkgo biloba* DXS1 AAS89341.1, DXS2 AAR95699.1, *Hevea brasiliensis* DXS1 AAS94123.1, DXS2 ABF18929.1, *Oryza sativa* DXS1 XP_015640505.1, DXS2 XP_015647944.1, DXS3 XP_015642490.1, *Pinus densiflora* DXS1 XP_015640505.1, DXS2 ACC54554, *Pinus taeda* DXS1 ACJ67021, DXS2 ACJ67020.1, *Pinus kesiya* DXS1 AIY22671.1, DXS2 AIY22670.1, *Populus trichocarpa* DXS1 XP_002312717.1, DXS2-1 XP_002303416.1, DXS2-2 XP_002331678.1, DXS3 XP_002308644.1, *Ricinus commol/Lunis* DXS1 XP_015573388.1, DXS2-1 XP_002532384.1, DXS2-2 XP_002533688, DXS3 XP_002514364, *Salvia miltiorrhiza* DXS1 ACF21004.1, DXS2 ACQ66107.1, *Vitis vinifera* DXS1XP_002277919.1, DXS2 XP_002266925.1, DXS3 XP_002282428.2, *Zea mays* DXS1 NP_001157805.1, DXS2 NP_001295426.1, DXS3 NP_001170088.1, *Escherichia coli* DXS1-1 EFF14228.1, DXS1-2 BAB33897.1, *Streptomyces cinereocoelicolor* DXS1 Q9X7W3, *Artemisia annua* DXS1 AAD56390.2, *Picea abies* DXS1 ABS50518.1, DXS2A ABS50520.1, DXS2B ABS50519.1).

**Figure 2 ijms-22-00848-f002:**
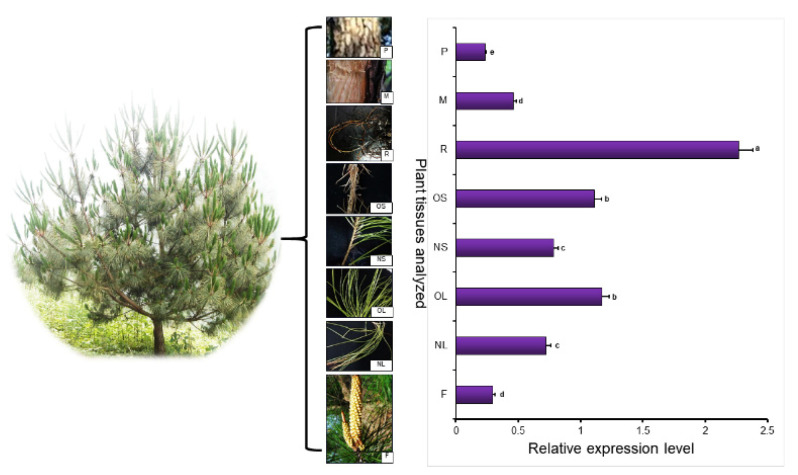
*PmDXS* expression level in different tissues of *P. massoniana*. The young leaf tissue was used as a control. Different lowercase letters in the same column indicate a significant difference, *p* < 0.05.

**Figure 3 ijms-22-00848-f003:**
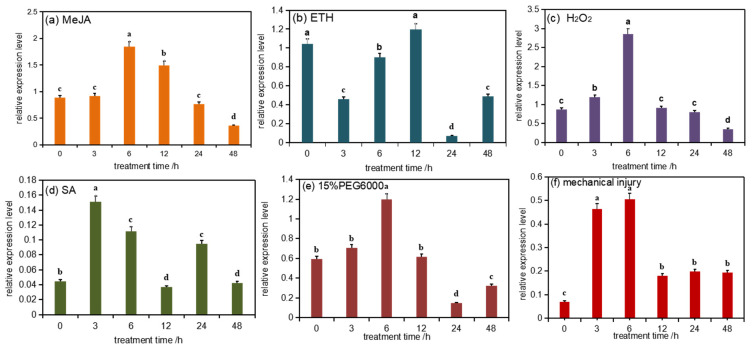
*PmDXS* expression level in different stress of *P. massonniana*. Different lowercases in the same column indicate a significant difference, *p* < 0.05. All the above treatments were treated at 0 h without any treatment as the control group. Figure (**a**) shows the relative expression level under MeJA treatment; figure (**b**) shows the relative expression under ETH treatment; figure (**c**) shows the relative expression under H_2_O_2_ treatment; figure (**d**) shows the relative expression under SA treatment; figure (**e**) shows the relative expression under 15 % PEG6000 treatment; figure (**f**) shows the relative expression under mechanical injury treatment.

**Figure 4 ijms-22-00848-f004:**
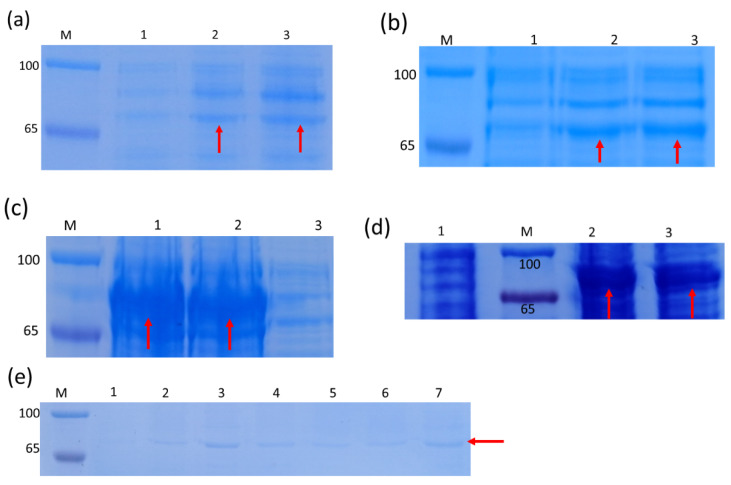
SDS-PAGE analysis of the expressed product of the recombinant strain pET28a-PmDXS. M. Protein molecular weight standard. The molecular weight of the target protein, which is about 70.4 kDa (red arrow). (**a**). The supernatant of bacterial solution at 16 °C (1. Recombinant strain without IPTG induction). 2. Recombinant strain with 0.5 mM IPTG induction for 16 h. 3. Recombinant strain with 1.0 mM IPTG induction for 16 h). (**b**). Induction of bacterial solution under 16 °C (1. Recombinant strain without IPTG induction; 2. Recombinant strain with 0.5 mM IPTG induction for 16 h; 3. Recombinant strain with 1.0 mM IPTG induction for 16 h). (**c**). The precipitate of the bacterial solution at 16 °C (1. Recombinant strain with 0.5 mM IPTG induction for 16 h; 2. Recombinant strain with 1.0 mM IPTG induction for 16 h; 3. Recombinant strain without IPTG induction). (**d**). Induction of bacterial solution under 37 °C (1. Recombinant strain without IPTG induction; 2. Recombinant strain with 0.5 mM IPTG induction for 4 h; 3. Recombinant strain with 1.0 mM IPTG induction for 4 h). (**e**). The supernatant of the bacterial solution at 37 °C (1. pET28a strain with 1.0 mM IPTG induction for 4 h; 2~7. Recombinant strain with 0, 0.1, 0.3, 0.5, 1.0, and 2.0 mM IPTG induction for 4 h).

**Figure 5 ijms-22-00848-f005:**
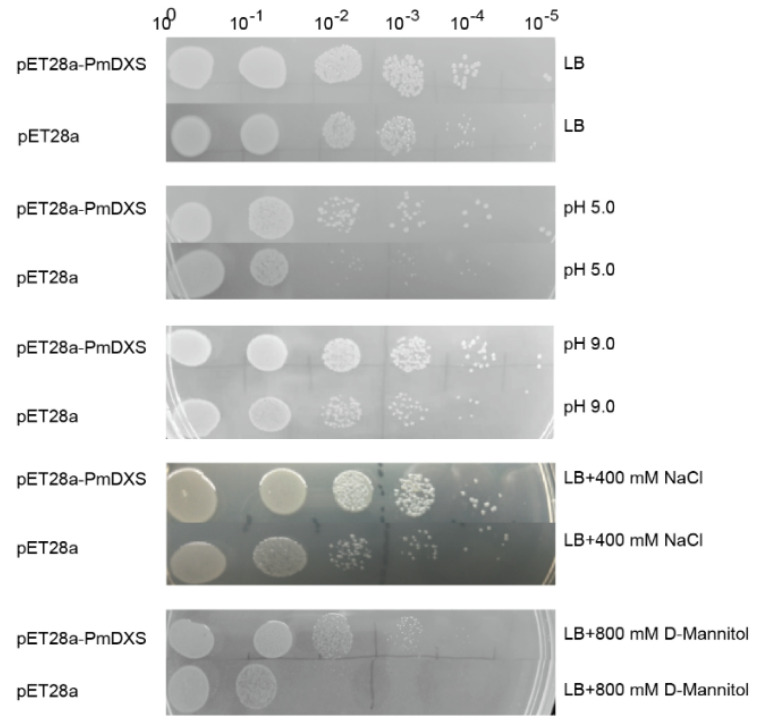
Growth status of recombinant and control bacteria on solid LB medium. The recombinant bacteria (TransB/pET28a−PmDXS) and the control bacteria (TransB/pET28a) were inoculated on solid plates containing LB, 400 mM NaCl, 800 mM D-mannitol, HCl (pH 5.0), or NaOH (pH 9.0). All plates were incubated at 37 °C overnight. 10^0^~10^−5^ represent the dilution gradients of the *E. coli* liquid culture.

**Figure 6 ijms-22-00848-f006:**
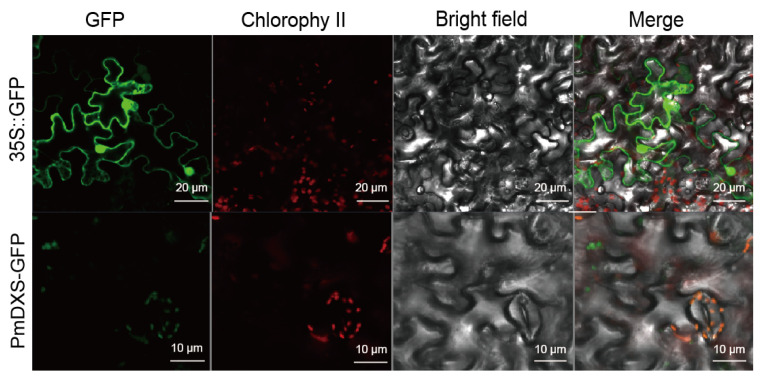
Subcellular localization of PmDXS in epidermal leaf cells of transiently infected *N. benthamiana*. Green is green fluorescent protein, red is chloroplast autofluorescence.

**Figure 7 ijms-22-00848-f007:**
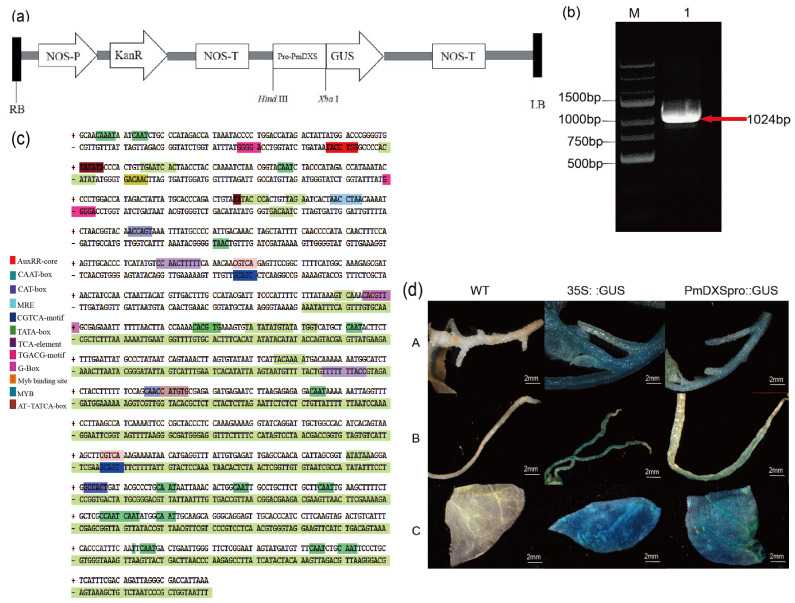
Analysis of the *PmDXS* promoter sequence and transient expression. (**a**) Schematic diagram of pBI121-PmDXSpro::GUS vector construction. (**b**) Cloning and verification of the *PmDXS* promoter from *P. massoniana*. M. DNA marker GsDL2502; 1. PCR amplification of *PmDXS* promoter (red arrow). (**c**) Nucleotide sequence and predicted cis-acting elements of the *PmDXS* promoter. (**d**) Detection of *PmDXS* promoter activity by transient expression in tobacco (A. Stems of tobacco B. Roots of tobacco C. Leaves of tobacco).

**Figure 8 ijms-22-00848-f008:**
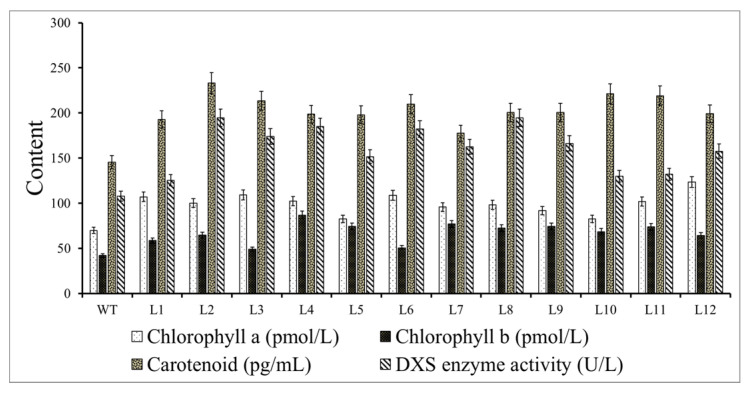
Contents of chlorophyll a, chlorophyll b, and carotenoids and 1-deoxy-D-xylose-5-phosphate synthase (DXS) enzyme activity in the transgenic lines and wild type (WT) *A. thaliana*.

**Table 1 ijms-22-00848-t001:** The sequences of primers used in this study.

Primer Name	Forward Sequence (5′–3′)	Reverse Sequence (5′–3′)
PmDXS-Mid	CCTCTGGTTTGGCTGGATTTCCC	CGGTGCTTCAGAAGAGCATCATG
PmDXS 5′ RACE	CCCGTTCAAAGCCAGGAAATGGGAGAC	ACCCTTGGCGGCTTCTCTGAGTTTGCG
PmDXS 3′ RACE	GGTTGGGGCAGACGGTCCTACTCATTG	TGGCTTTGAACGGGCTACTTGATGGAA
DXS-ORF	ATGGCAATTGCAAGCAGGGCAGGAG	TTATCGGTGCTTCAGAAGAGCATC
Actin2	CACGGAATAGGCAGAAGTTGG	TGGGCATAAAGTGTTAGAATAGC
Q-DXS	CAGTCTGCAATACCCTGCTCAT	CTGCTTTCACATTTTTCCCCTT
28a-DXS	gggtcgcggatccgaattcGCAATTGCAAGCAGGGCAGGAGT ^1^	caagcttgtcgacggagctcTCGGTGCTTCAGAAGAGCATCA ^1^
DXS-GFP	gagaacacgggggactctagaATGGCAATTGCAAGCAGGG ^1^	gcccttgctcaccatggatccTCGGTGCTTCAGAAGAGCATC ^1^
GSP1		TAATGGTCGCCCTAATCTGTCG
GSP2		CGCTAATGTGTTGGCTCAATCTCA
1302-DXS	acgggggactcttgaccatggATGGCAATTGCAAGCAGGG ^1^	aagttcttctcctttactagttTATCGGTGCTTCAGAAGAG^1^
PmDXS-Pro	GCAACAAATAATCAATCTGCCCATAG	GAAATGAGCAGGGAATTGCAGATT
pBI121-DXS	caaagggcaatcgggggactGCAACAAATAATCAATCTGCCCATAG ^1^	cgtaacataagggactgaccacGAAATGAGCAGGGAATTGCAGATT ^1^
pBI121	GGTGCTACTGGTGATTTTGCTG	CTGATGCTCCATCACTTCCTG

^1^ Lowercase letters indicate the sequence of the expression vector, and uppercase letters indicate the sequence of the insert.

## Data Availability

Not applicable.

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
