# Peer review of "Characterization and Function of the 1-Deoxy-D-xylose-5-Phosphate Synthase (DXS) Gene Related to Terpenoid Synthesis in Pinus massoniana"

_ijms, 2021, doi:10.3390/ijms22020848_

Round 1
Reviewer 1 Report
Abstract
1-deoxy-d-xylose => 1-deoxy-D-xylose
"with PmDXS proteins from other species" => rather "with DXS proteins from other species"
Major
1. I think Figure 1 needs improvements
- resolution of most subfigures
- subfigure "D" label not visible and same height as rest
- many figures look stretched, e.g. D
- in F) explain red/green rectangles
- itol diagrams
itol offers possibilites to add coloring features, in order to
make G more understandable and relevant add taxonomic infomation
2. You should explain in MS Figure 3:
- why 6hours later maximum
- why SA already maximum at 3 h
Material and Methods:
1. why 2- and 15-year old masson pine?
2. Table-1, why upper- and lower-case?
3. Check whether contains all references to original studies of the tools listed in the manuscript
Results
1. PLN02582
2. GDGG(X)8 => GDGG(X){8} ?
3. Figure 2: order the tissues from heighest relative expression to lowest,
better: use a schematic illustration of P. massoniana where you place the expression to the plant parts
y-axis: what is the unit of "relative expression level"
4. 22 sites TATA and CAAT (22 vs 24): is that more than expected?
Author Response
Response to Reviewer 1 Comments
Dear Editor and Reviewers:
Manuscript ID: ijms-1049861.
Type: research article.
Manuscript Title: Characterization and Function of the 1-Deoxy-D-xylose-5-Phosphate Synthase (DXS) Gene Related to Terpenoid Synthesis in Pinus massoniana
We are very grateful to Reviewer for reviewing the paper so carefully. We have carefully considered the suggestion of Reviewer and make some changes. We used the "Track Changes" function in Microsoft Word, so that changes are easily visible to the editors and reviewers.
Point 1: Abstract
1-deoxy-d-xylose => 1-deoxy-D-xylose
Response 1: We appreciate it very much for this good suggestion, and we have done it according to your ideas. We are very sorry for our incorrect writing and it is rectified. Line12: 1-deoxy-d-xylose.
Point 2: "with PmDXS proteins from other species" => rather "with DXS proteins from other species"
Response 2: Thanks for your kind correction. It is rectified at Line14(with DXS proteins from other species)
Point 3: Major
1.I think Figure 1 needs improvements
- resolution of most subfigures.
Response 3: Thank you for your suggestion. We made improvements based on your suggestions. We have carefully revised Figure 1 and improved the clarity of all subfigures.
Point 4: - subfigure "D" label not visible and same height as rest.
Response 4: According to your suggestions, we have unified the overall picture in Figure 1. We changed the label of the picture to superscript lowercase letters to make it easier to see.
Point 5:- many figures look stretched, e.g. D.
Response 5: Figure D is a perspective view, not a plan view. It may feel stretched. According to your suggestions, We have revised Figure D again.
Point 6: - in F) explain red/green rectangles
Response 6: It is really a good idea as Reviewer suggested, and we have changed them all to meet Reviewer’s thoughts. We are very sorry for our negligence of the explanation. Based on your suggestion, we explained the red/green rectangles in Figure F.
Line 248: The "jnetpred" in the picture represents the prediction result of the secondary structure of DXS protein: α-helices is represented by red tubes, and β-sheets is represented by green arrows.
Point 7: - itol diagrams
itol offers possibilites to add coloring features, in order to make G more understandable and relevant add taxonomic information
Response 7: It is really a great suggestion as Reviewer pointed out that itol offers possibilites to add coloring features, in order to make G more understandable and relevant add taxonomic information. We have marked the molecular phylogenetic tree with multiple colors, Line 253.
Point 8: You should explain in MS Figure 3
- why 6 hours later maximum. - why SA already maximum at 3 h.
Response 8: We are very grateful to Reviewer for this idea. We made some explanations in the discussion section, Line 443-452. We speculate that under different treatments, the response rate of PmDXS gene to different adversities is different, so the time point is different.
Line 443-452: This study also confirmed that treatment with MeJA (10 mM), H2O2 (10 mM), SA (1 mM), ETH, and PFG6000 (15%) induced upregulated expression of PmDXS. In addition, PmDXS showed a trend of first rising, then falling and then rising after mechanical damage treatment. PmDXS was upregulated and expressed in a short period of time, suggesting that this gene may be involved in the synthesis of terpenoids under adverse stress, thereby resisting the damage to Masson pine caused by adverse factors. Therefore, it can be speculated that the DXS gene has temporal and spatial specificity, which can increase the expression of PmDXS at the transcription level, thereby affecting the synthesis of terpenoids and improving stress resistance. This agrees with research on the PdDXS gene of P. densiflora and the GbDXS gene of Ginkgo biloba.
Point 9: Material and Methods:
- why 2- and 15-year old masson pine?
Response 9: The suggestion you mentioned is really good. When we were doing experiments in the early stage, we needed different organizations of masson pine (especially the organization of flowers). We consider that the 2-year-old masson pine does not have flowers, and there are not enough xylem samples and phloem samples for our experiments. Therefore, we chose 15-year-old masson pine for the relative expression test of different tissues.In fact, the main reason is that 15-year-old masson pine is relatively easy to obtain when we take samples (Xylem and Phloem). In addition, We consider that when using 2-year-old potted seedlings for different stress treatments, it is easier to collect the needles of Masson pine every 0, 3, 6, 12, 24, and 48 h.
Point 10: 2. Table-1, why upper- and lower-case?
Response 10: We are very grateful for your suggestion. We made changes based on your comments. Explained below the table, Line106. When we construct the expression vector, lowercase letters indicate the sequence of the expression vector, and uppercase letters indicate the sequence of the inserted fragment.
Point 11: 3. Check whether contains all references to original studies of the tools listed in the manuscript.
Response 11: It is really a great suggestion. Considering the Reviewer’s suggestion, we have added references to original studies of the tools listed in the manuscript, Line 558-567.
Line 558-567: 13.Alexey, D.; Christin, C.; James, P.; et al. JPred4: a protein secondery structure prediction server. Nucleic Acids Research 2015, 43(W1): w389-w394. DOI: 10.1093/nar/gkv332.
14.Kumar, S.; Stecher, G.; Li, M.; et al. MEGA X: Molecular evolutionary genetics analysis across comuting platforms. Molecular Biology & Evolution 2018, 1-8. DOI: 10.1093/molbev/msy096.
15.He, Z. L.; Zhan, H. K.; Gao, S. H.; et al. Evolview v2: an online visualization and management tool for customized and annotated phylogenetic trees. Nucleic Acids Research 2016, 44: W236-W241. DOI: 10.1093/narlgkw370.
16.Chen, K. C.; Shen, H. B. Cell-PLoc 2.0: an improved package of web-servers for predicting subcellular localization of proteins in various organisms, Natural Science 2010, 2: 1090-1103. DOI: 10.4236/n.
Point 12: Results
1.PLN02582
Response 12: There is a multifunctional domain of PLN02582, which is a characteristic sequence of 1-deoxyxylulose-5-phosphate synthase.
Line 227: Between the 73rd and 736th amino acids, there is a multifunctional domain of PLN02582, which is a characteristic sequence of 1-deoxyxylulose-5-phosphate synthase.
Point 13: 2.GDGG(X)8 => GDGG(X){8} ?
Response 13: Thanks for your kind suggestions. The PmDXS protein was found that it contained the ammonium disulfate binding site GDGG(X)8E(X)4A(X)11NDN (marked with red dotted line) and a transketolase domain DRAGX28PXD (marked with red solid line), Line230.
Point 14: 3.Figure 2: order the tissues from heighest relative expression to lowest,better: use a schematic illustration of P. massoniana where you place the expression to the plant part.
Response 14: Thanks very much for your comment, which is highly appreciated. According to your suggestions, We have modified Figure 2 to add a schematic diagram of the different organizations of Masson pine, and corresponded to it one by one. We have modified Figure 2 to add a schematic diagram of the different organizations of Masson pine, and corresponded to it one by one. Please look at the picture below.
|
Point 15: 4.y-axis: what is the unit of "relative expression level"y.
Response 15: We carefully considered your kind suggestion, however, what we did was a relative expression experiment. All the values we obtained in the experiment are compared with the reference value (CK), so the values of other groups are relative values.
Point 16: 5.22 sites TATA and CAAT (22 vs 24): is that more than expected?
Response 16: Thank you very much for your question. CAAT-box (common cis-acting element in promoter and enhancer regions)、TATA-box (core promoter element around -30 of transcription start). PmDXS promoter sequence contained 10 types of cis-acting elements.What we want to explain in this article is that TATA and CAAT are typical core elements of eukaryotic promoters. The 22 and 24 in the paper only represent their respective numbers and have no special meaning for comparison.
We would like to thank the referee again for taking the time to review our manuscript.

Reviewer 2 Report
The manuscript by Li et al., investigated the role of 1-Deoxy-D-xylose-5-Phosphate Synthase (PmDXS) gene in Masson pine terpenoid biosynthesis. The study isolated the full CDS of PmDXS and did an amino acid alignments and phylogenetic analysis in various related species. They performed tissue specific relative expression of PmDXS in various tissue types to multiple abiotic stresses to find the relationship of PmDXS expression to a particular stress environment. They also expressed the recombinant protein in E.coli, however not clear on how this expressed protein has been used? Recombinant bacterial growth under salt stress has been shown to be better than empty vector. Sub-cellular localization of the gene has been shown to be in chloroplast. The promoter of the gene has been amplified and analyzed for the presence of regulatory elements and its function tested by GUS expression. The chlorophyll and carotenoid contents were analyzed were higher in Arabidopsis PmDXS over expression lines than control. This study is essentially about the PmDXS gene and its function and expression patterns.
Line#34: haploids? This seems to be an error. IPP and DMAPP are synthesized in MVA pathway. These precursors are condensed through sequential addition to generate larger isoprenoid precursor molecules.
Line#56: Is DXS over expressed in Zanthoxylum as well and relative expression levels of genes increased? That has to me made clear for the readers.
Line#95: The correct band was extracted from the gel, cloned, and sequenced? Or the PCR products were directly sent for sequencing. Method is not clearly written. IF cloned into a bacterial vector, details must be provided.
Line#191: Floral dip method instead of inflorescence infection
Line#212: Some of the results section is more like materials and methods and the relevance of some of the results is unclear.
Line#256: This sentence can be in the figure legend. Figure#2 x-axis label could be “Plant tissues analyzed”
Line#267: incomplete sentence?
Line#352: Figure#7 Gel image doesn’t stand by itself as a figure on its own. It could probably be combined with another figure.
Author Response
Response to Reviewer 2 Comments
Dear Editor and Reviewers:
Manuscript ID: ijms-1049861.
Type: research article.
Manuscript Title: Characterization and Function of the 1-Deoxy-D-xylose-5-Phosphate Synthase (DXS) Gene Related to Terpenoid Synthesis in Pinus massoniana
We are very grateful to Reviewer for reviewing the paper so carefully. We have carefully considered the suggestion of Reviewer and make some changes. We used the "Track Changes" function in Microsoft Word, so that changes are easily visible to the Editors and Reviewers.
Point 1: Comments and Suggestions for Authors: They also expressed the recombinant protein in E.coli, however not clear on how this expressed protein has been used?
Response 1: Thank you for this valuable question. The purpose of our prokaryotic expression experiment is to verify whether the protein encoded by the PmDXS gene is consistent with the predicted size (70.4 KDa). However, whether the protein can be induced is affected by many factors, such as: environmental temperature, IPTG concentration, method of breaking, and so on. In this study, we used 37°C and 16°C (that is, higher temperature and lower temperature) as environmental temperature variables, and found that the amount of protein induced at 37°C was larger, and the protein size was consistent with that predicted by the website (70.4 KDa). Subsequently, we used this temperature as a non-variable and designed 6 IPTG concentrations (0, 0.1, 0.3, 0.5, 1.0, and 2.0 mM IPTG induction for 4 h) for protein induction. We found that there was no significant difference in protein expression under different IPTG concentrations. Finally, we designed 4 abiotic stresses (400 mM NaCl, 800 mM D-mannitol, pH 5.0 (HCl), or pH 9.0 (NaOH)) to verify the stress response ability of the recombinant bacterial solution. We found that pET28a-PmDXS recombinant bacteria have the ability to positively regulate abiotic stress in plants. This is similar to the findings of Populus trichocarpa [1].
[1] Wei. H.; Movahedi. A.; Xu. C.; et al. Overexpression of PtDXS enhances stress resistance in poplars. International Journal of Molecular Sciences 2019, 20(7): 1669-1690. DOI: 10.3390/ijms20071669.
Point 2: haploids? This seems to be an error. IPP and DMAPP are synthesized in MVA pathway. These precursors are condensed through sequential addition to generate larger isoprenoid precursor molecules.
Response 2: Thanks for your kind suggestions,which is valuable for improving the accuracy of the manuscript. We are very sorry for the wrong description of the word. We have made a modification, monoterpenes are the main type of terpenoids found. Line35. In addition, the main products of the MVA pathway are monoterpenoids and diterpenes, while the MEP pathway can synthesize sesquiterpenes.
Point 3: Line#56: Is DXS over expressed in Zanthoxylum as well and relative expression levels of genes increased? That has to me made clear for the readers.
Response 3: Thanks for your constructive suggestion, which is highly appreciated. In this article: In this study, they used two variants of Zanthoxylum bungeanum as experimental materials. They analyzed volatile organic compounds in Z. bungeanum by HS-SPME-GC-MS. They designed primers based on conservative sequence alignment and passed RT -qPCR studies the expression profiles of candidate genes in metabolic pathways. Subsequently, they combined these results with bioinformatics to analyze the key genes identified. They found that the key rate-limiting enzyme ZbDXS in the MEP pathway showed the strongest upregulation in gene transcript level [2].
[2]Shi, J.; Fei, X.; Hu, Y.; et al. Identification of key genes in the synthesis pathway of volatile terpenoids in fruit of Zanthoxylum bungeanum Maxim. Forests, 2019, 10, 328. DOI: 10.3390/f10040328.
Point 4: Line#95: The correct band was extracted from the gel, cloned, and sequenced? Or the PCR products were directly sent for sequencing. Method is not clearly written. IF cloned into a bacterial vector, details must be provided.
Response 4: Thank you very much for your kind suggestion. We revised this description again.
Line 96: After gel electrophoresis detection, we recover of the products from the gel according to the manufacturer's instructions (Yeasen Biotech, Shanghai, China). The recovered product was ligated into pEASY-Blunt vector (Transgen Biotech, Beijing, China), transformed into Escherichia coli Trans1-T1 (Transgen Biotech, Beijing, China), and cultured at 37℃ overnight. Blue and white spots were screened with LB solid medium containing X-Gal and IPTG. Then a single white colony was selected for PCR detection, and the recombinant plasmid was sequenced by Jie Li Company (Shanghai, China).
Point 5: Floral dip method instead of inflorescence infection
Response 5: Thanks for your constructive suggestion.We have carefully revised this, Line 200(Floral dip).
Point 6: Line#212: Some of the results section is more like materials and methods and the relevance of some of the results is unclear.
Response 6: Thank you for your kind suggestions. After considering your suggestion, we reorganized the expression of the sentence. In addition, we streamlined some redundant conclusions to make our articles more intuitive and understandable.
Line 219: Similar to DXS proteins in other plants, the PmDXS protein is a hydrophilic protein and it does not contain a transmembrane domain and signal peptide.
Point 7: Line#256: This sentence can be in the figure legend. Figure#2 x-axis label could be “Plant tissues analyzed”.
Response 7: Thanks for your suggestions. According to your suggestions, we have improved the overall effect of Figure 2 and combined the Relative expression level with the schematic diagram of Masson pine.
Point 8: Line#267: incomplete sentence?
Response 8: Thank you very much for the positive comments and constructive suggestions. Based on your suggestions, we made changes to the results description in this section.
Line 285: qRT-PCR analysis showed that the relative expression levels of PmDXS under the conditions of mechanical damage with 15% PEG 6000, 10 mM H2O2, 50 μM ETH, 10 mM MeJA, and 1 mM SA were significantly higher than those of the control group ,Figure 3(a)-(f).
Point 9: Line#352: Figure#7 Gel image doesn’t stand by itself as a figure on its own. It could probably be combined with another figure.
Response 9: We are very grateful for your suggestions for this picture. We are very grateful for your suggestions for this picture. We have combined Figure 7 with Figure 8.
We would like to thank the referee again for taking the time to review our manuscript.

Round 2
Reviewer 1 Report
Dear authors,
thanks for improving the figures and efforts.
- Figure 1 is now much improved, still you might try to put the labels with same
font size and font height.
p4/l135: Actin2 or Actin?, probably recommended name Actin-2?
https://www.uniprot.org/uniprot/Q96292
Author Response
Response to Reviewer 1 Comments
Dear Editor and Reviewers:
Manuscript ID: ijms-1049861.
Type: research article.
Manuscript Title: Characterization and Function of the 1-Deoxy-D-xylose-5-Phosphate Synthase (DXS) Gene Related to Terpenoid Synthesis in Pinus massoniana.
We are very grateful to Reviewer for reviewing the paper so carefully. We have carefully considered the suggestion of Reviewer and make some changes. We used the "Track Changes" function in Microsoft Word, so that changes are easily visible to the editors and reviewers.
Point 1:Figure 1 is now much improved, still you might try to put the labels with same font size and font height.
Response 1:Thanks for your kind suggestions. According to your suggestion, we standardized all the label letters with same font size and font height in Figure 1.
Point 2:p4/l135: Actin2 or Actin?, probably recommended name Actin-2?
Response 2:All of us think that your suggestion is very helpful in improving our article. We are sorry that forgot to mark the NCBI accession number. This is indeed a small mistake in our writing process. We have marked the NCBI accession number and the source of the original research references on Line135 Actin2(NCBI accession number: KM496525.1) , so that readers can better understand our research. Actin2 is a reference gene of Pinus massoniana.
